# Changes in Gut Microbiota According to Disease Severity in a Lupus Mouse Model

**DOI:** 10.3390/ijms26031006

**Published:** 2025-01-24

**Authors:** Eui-Jeong Han, Ji-Seon Ahn, Yu-Jin Choi, Da-Hye Kim, Hea-Jong Chung

**Affiliations:** Gwangju Center, Korea Basic Science Institute, Gwangju 61751, Republic of Korea; iosu5772@kbsi.re.kr (E.-J.H.); ajs0105@kbsi.re.kr (J.-S.A.); cyj4854@kbsi.re.kr (Y.-J.C.); rlaek9925@kbsi.re.kr (D.-H.K.)

**Keywords:** lupus pathogenesis, gut microbiota, immune dysregulation, biomarkers

## Abstract

Systemic lupus erythematosus (SLE) is a multifaceted autoimmune disease driven by immune dysregulation. This study investigated the relationship between gut microbiota and lupus severity using the MRL/lpr lupus mouse model. Mice were grouped based on total immunoglobulin (Ig)G, IgG2a levels, and urine albumin-to-creatinine ratio (ACR), allowing for the comparison of gut microbiota profiles across different disease severities. Interestingly, severe lupus mice exhibited significant reductions in *Ruminiclostridium cellulolyticum*, *Lactobacillus johnsonii*, and *Kineothrix alysoides*, while *Clostridium saudiense*, *Pseudoflavonifractor phocaeensis*, and *Intestinimonas butyriciproducens* were enriched. These microbial shifts correlated with elevated IgG, IgG2a, and ACR levels, indicating that changes in the gut microbiome may directly influence key immunological markers associated with lupus severity. The depletion of beneficial species and the enrichment of potentially pathogenic bacteria appear to contribute to immune activation and disease progression. This study suggests that gut microbiota dysbiosis plays a critical role in exacerbating lupus by modulating immune responses, reinforcing the link between microbial composition and lupus pathogenesis. Our findings provide the first evidence identifying these distinct gut microbial species as potential contributors to lupus severity, highlighting their role as key factors in disease progression.

## 1. Introduction

Systemic lupus erythematosus (SLE) is a chronic autoimmune disease with diverse clinical manifestations, ranging from mild fatigue and joint pain to severe complications such as lupus nephritis and neuropsychiatric disorders [1]. The pathogenesis of SLE is complex and involves a multifactorial interplay of genetic, environmental, and immunological factors [2]. Genetic predispositions, including variations in immune-regulating genes (HLA-DR2/DR3, IRF5, and STAT4), contribute to immune dysregulation, defective clearance of apoptotic cells, and activation of type I interferon pathways [3]. Environmental triggers, such as UV radiation, infections, hormonal changes, and gut microbiota dysbiosis, further influence disease onset and progression [4]. These combined factors result in the loss of self-tolerance, production of autoantibodies, immune complex deposition, and chronic inflammation, which are hallmarks of SLE pathophysiology [4].

Animal models are essential for studying SLE pathophysiology, with the MRL/lpr mouse model being one of the most widely used [5]. This model exhibits key features of human lupus, including the development of lupus nephritis, autoantibody production, and systemic inflammation [5,6]. The *lpr* mutation in the *Fas* gene of these mice leads to defective apoptosis and aberrant lymphoproliferation, making it a valuable tool for investigating the mechanisms underlying lupus pathogenesis [7]. Lupus disease onset generally occurs around 16 weeks of age in these mice. However, since biomarkers do not fully capture the entire spectrum of the disease, it is important to note that not all 16-week-old mice exhibit the same level of lupus severity [8].

Currently, pharmacological treatments for lupus include corticosteroids, immunosuppressants (e.g., cyclophosphamide, mycophenolate mofetil), and antimalarials (e.g., hydroxychloroquine), which manage symptoms and prevent disease flares [9]. However, these treatments are often associated with significant side effects, including immunosuppression, increased infection risk, and organ toxicity [10]. As a result, there is growing interest in exploring alternative therapeutic strategies, including microbiota-based interventions and novel biomolecules that may offer more targeted and less toxic options.

Recent studies indicate that gut microbiota plays a pivotal role in maintaining immune homeostasis and influencing the pathogenesis of autoimmune diseases, including SLE [11]. Dysbiosis—marked by an overgrowth of pathobionts such as *Ruminococcus gnavus* and a reduction in short-chain fatty acid-producing bacteria—has been associated with heightened systemic inflammation and increased lupus severity [12,13]. This highlights the urgent need for comprehensive microbial profiling and characterization to better understand the relationship between gut microbiota and autoimmune pathogenesis.

While these findings underscore the potential involvement of the gut microbiome in lupus progression, research identifying the specific microbial species linked to lupus and other autoimmune diseases remains limited. Despite growing recognition of the microbiota’s influence on autoimmunity, few studies have focused on distinguishing the particular microbial taxa that drive disease progression and severity. This highlights the urgent need for comprehensive microbial profiling and characterization to better understand the relationship between gut microbiota and autoimmune pathogenesis.

In this study, we classified 16-week-old mice based on lupus severity and compared lupus-related biomarkers between these groups and normal controls. Specifically, we aimed to identify biomarkers that can assess lupus severity in animal models and explore the gut microbiota’s potential influence on these biomarkers, contributing to disease severity. Our findings provide new insights into the relationship between biomarkers, gut microbiota, and lupus, emphasizing the importance of microbial species identification and their potential implications for developing targeted interventions.

## 2. Results

### 2.1. Grouping Based on Immunological Markers of Lupus

Lupus-prone mice were grouped based on albumin-to-creatinine ratio (ACR), total IgG, and IgG2a levels, which are well-established biomarkers of lupus severity (Table 1). At 6 weeks of age, five randomly selected mice with no visible signs of disease exhibited low levels of IgG (210.55 ± 44.92 µg/mL), IgG2a (1013.70 ± 30.26 µg/mL), and ACR (44.51 ± 12.14 mg/g), representing the Lupus-weak group. At 16 weeks of age, lupus-prone mice displayed varying degrees of biomarker elevation and were categorized into two groups—Lupus-medium, and Lupus-strong—according to the distribution of IgG, IgG2a, and ACR levels. Mice with moderate biomarker levels were assigned to the Lupus-medium group, characterized by IgG levels of 2069.26 ± 879.29 µg/mL, IgG2a levels of 1691.75 ± 9.54 µg/mL, and ACR values of 102.34 ± 20.25 mg/g. Mice exhibiting the highest biomarker levels were categorized as Lupus-strong, with IgG (3133.95 ± 649.04 µg/mL), IgG2a (1736.46 ± 21.34 µg/mL), and ACR (461.16 ± 188.68 mg/g), indicating more severe disease and greater renal impairment. These findings highlight that differences in lupus severity are reflected by variations in ACR, IgG, and IgG2a levels, reinforcing previous studies that demonstrate the correlation between biomarker levels and disease progression.

### 2.2. Progressive Changes in Fecal Gut Microbiota Composition and Diversity with Lupus Severity

Figure 1 illustrated the differences in gut microbiota composition and diversity among fecal samples collected from lupus-affected mice, which were classified by disease severity into mild (Fecal_W, red), moderate (Fecal_M, green), and severe (Fecal_S, blue) groups.

Figure 1A compared the alpha diversity of the three groups using Fisher and Simpson indices. The Fecal_W group exhibited the lowest median diversity for both indices, indicating reduced species richness and evenness in mice with mild lupus symptoms. In contrast, the severe lupus group showed higher species richness and microbial diversity compared to the mild group. This unexpected increase in microbial diversity may reflect a disruption of gut homeostasis driven by disease severity, potentially leading to microbial dysbiosis rather than a stable, healthy community. The moderate lupus group displayed a higher median Fisher index and greater variability, suggesting increased species richness with some fluctuation in community composition, further supporting the notion that microbiota restructuring occurs progressively as lupus severity advances.

Figure 1B showed the relative abundance of gut microbial species across the three groups. In the mild lupus group, species such as *Eubacterium siraeum* and *Oscillibacter valericigenes* were more prevalent. The moderate lupus group exhibited a more balanced distribution, with notable contributions from *Anaerotignum lactatifermentans* and *Christensenella timonensis*. In the severe lupus group, *Alistipes putredinis* and *Clostridium scindens* dominated, indicating a progressive shift in microbiota composition as disease severity increased.

Figure 1C,D highlighted compositional differences and clustering of microbial communities. The multidimensional scaling (MDS) plot (Figure 1C) showed clear separation between the three groups, indicating distinct microbial profiles that diverged with disease severity. The heatmap (Figure 1D) revealed patterns of microbial species abundance, with hierarchical clustering reflecting group-specific microbial signatures. Enrichment and depletion of certain taxa correlated with lupus severity, suggesting microbiota restructuring during disease progression.

Figure 1E,F further characterize microbial community structures and taxonomic compositions. Figure 1E presents microbial co-occurrence networks, highlighting differences in complexity and connectivity between groups, with more intricate and interconnected networks observed in severe lupus. Figure 1F shows phylogenetic trees, illustrating shifts in microbial lineages, with notable taxonomic diversity and reorganization in severe cases. The colors representing phylum and family levels in Figure 1F are detailed in Appendix A.

Collectively, these figures illustrate the progressive alteration of gut microbiota in lupus-affected mice, highlighting microbial diversity, composition, and network dynamics that align with disease severity.

### 2.3. Progressive and Detailed Characterization of Gastrointestinal (GI) Microbiota Reveals Greater Microbial Shifts with Lupus Severity Compared to Fecal Analysis

Figure 2 illustrates the gut microbiota composition and diversity within the gastrointestinal (GI) tract of lupus-affected mice, categorized by disease severity. Figure 2A presented the alpha diversity of GI tract microbial communities, serving to cross-validate the findings observed in Figure 1A. Consistent with fecal microbiota analysis, the Fisher and Simpson indices revealed that microbial diversity and evenness increased with lupus severity. The mild lupus group (red) exhibited the lowest diversity, while the severe group (blue) showed the highest values, suggesting a parallel trend of rising species richness within the GI tract. This increase in microbial diversity, observed both in fecal and GI tract samples, likely reflects the disruption of gut homeostasis driven by disease severity, fostering microbial dysbiosis and the potential proliferation of pathogenic bacteria as lupus progresses.

Figure 2B shows the relative abundance of microbial species in the GI tract across disease severity groups. Compared to fecal samples in Figure 1B, the GI tract harbored greater microbial diversity and species richness, indicating a more complex and heterogeneous microbial environment. Key species such as *Clostridium scindens*, *Eisenbergiella massiliensis*, and *Oscillibacter valericigenes* appeared across all severity groups. Additionally, taxa including *Lactobacillus reuteri*, *Flintibacter butyricus*, and *Roseburia faecis* contributed to the GI microbial landscape, underscoring the broader taxonomic diversity within the GI tract.

Figure 2C displays a multidimensional scaling (MDS) plot of GI tract microbial communities, revealing distinct clustering by disease severity. The separation between mild (GI_W, red), moderate (GI_M, green), and severe (GI_S, blue) groups was more pronounced in the GI tract compared to fecal samples, suggesting localized microbial shifts that might not have been fully reflected in fecal analyses.

Figure 2D presented a heatmap of microbial species abundance in the GI tract. Compared to Figure 1D, the GI tract showed broader species representation and clearer differentiation between severity groups. The hierarchical clustering highlighted distinct patterns of enrichment and depletion, reflecting a more variable microbial composition that aligned closely with lupus severity.

Figure 2E illustrates microbial co-occurrence networks in the GI tract. The networks showed greater connectivity and larger clusters than those observed in fecal samples (Figure 1E), suggesting more intricate and localized microbial interactions within the GI environment. This complexity might have played a significant role in disease progression.

Figure 2F presents phylogenetic trees representing the taxonomic composition of GI tract microbial communities. The GI tract exhibited greater phylogenetic diversity and pronounced shifts in taxonomic distribution across severity levels. This reflected a richer and more dynamic microbial ecosystem compared to fecal microbiota (Figure 1F). The colors representing phylum and family levels in Figure 2F are detailed in Appendix A.

Collectively, Figure 2 highlighted the greater complexity, diversity, and connectivity of GI tract microbiota in lupus-affected mice. These differences suggested that microbial alterations within the GI tract were more extensive and localized, which might have influenced microbial dysbiosis by contributing to the proliferation of harmful bacteria and exacerbation of lupus symptoms. This indicated that analyzing the GI tract could provide a more accurate reflection of microbial changes associated with lupus severity compared to fecal analyses.

### 2.4. Fecal Microbiota Analysis Reveals Key Microbial Shifts Associated with Lupus Severity Through DESeq and LEfSe Approaches

Figure 3 illustrates the differences in fecal microbiota composition between lupus-affected mice categorized by disease severity, utilizing DESeq and LEfSe analyses to identify differentially abundant microbial species. Microbial abundance was calculated using abundance values to reflect the differences in species richness. Figure 3A presents the results of the DESeq analysis, highlighting significant microbial taxa that differed between the mild and severe lupus groups. Key taxa, including *Clostridium saudiense*, *Turicibacter sanguinis*, and *Ligilactobacillus animalis*, were enriched in the severe lupus group, while species such as *Hoylesella oralis*, *Phocaeicola sartorii*, and *Eisenbergiella massiliensis* were more abundant in the mild group. The analysis demonstrated that Firmicutes, particularly *Clostridium saudiense* and *Turicibacter sanguinis*, were predominant in the severe group, whereas certain Bacteroidetes were enriched in the mild group.

Figure 3B shows the LEfSe analysis results, which similarly identified differentially enriched taxa between the mild and severe lupus groups. Species such as *Clostridium saudiense*, *Turicibacter sanguinis*, and *Ligilactobacillus animalis* were enriched in the severe lupus group, aligning with the findings from DESeq analysis. Conversely, *Hoylesella oralis*, *Phocaeicola sartorii*, and *Eisenbergiella massiliensis* were enriched in the mild group, reinforcing the overlap with taxa identified in Figure 3A. LEfSe results confirmed the predominance of Firmicutes in the severe group and Bacteroidetes in the mild group.

A notable overlap between the DESeq and LEfSe analyses was observed, with taxa including *Clostridium saudiense*, *Turicibacter sanguinis*, and *Ligilactobacillus animalis* consistently enriched in the severe lupus group. In contrast, *Hoylesella oralis*, *Phocaeicola sartorii*, and *Eisenbergiella massiliensis* were consistently associated with the mild lupus group. These overlapping taxa were visualized in Figure 3C, which graphically represents the dominant species shared between the two analyses. The consistent identification of these taxa underscores their potential role in lupus progression and their relevance as biomarkers for disease severity.

Additionally, we performed DESeq and LEfSe analyses using read counts rather than abundance values, which yielded similar results (Appendix A).

Figure 3C display correlation analyses, illustrating the relationship between the relative abundance of key microbial species and lupus severity. Strong positive correlations (R^2^ values) were observed for *Clostridium saudiense*, *Turicibacter sanguinis*, and *Ligilactobacillus animalis*, taxa that were consistently enriched in the severe group. Conversely, *Hoylesella oralis*, *Phocaeicola sartorii*, and *Eisenbergiella massiliensis* exhibited negative correlations and were more abundant in the mild lupus group.

### 2.5. Gastrointestinal Tract Microbiota Shows Stronger Association with Lupus Severity and Provides Greater Sensitivity and Accuracy Compared to Fecal Analysis

Figure 4 illustrates the differences in GI tract microbiota composition between lupus-affected mice categorized by disease severity, utilizing DESeq and LEfSe analyses to identify differentially abundant microbial species. Microbial abundance was calculated using abundance values to reflect differences in species richness. Figure 4A presents the results of the DESeq analysis, identifying significant microbial taxa that differed between the mild and severe lupus groups. Notable taxa, including *Clostridium saudiense*, *Pseudoflavonifractor phocaeensis*, and *Intestinimonas butyriciproducens*, were enriched in the severe lupus group, while taxa such as *Kineothrix alysoides*, *Lactobacillus johnsonii*, and *Ruminiclostridium cellulolyticum* were more abundant in the mild group. The analysis highlights that Firmicutes, particularly *Clostridium saudiense* and *Pseudoflavonifractor phocaeensis*, were dominant in the severe group.

Figure 4B shows the results of the LEfSe analysis, which further identified differentially enriched taxa between the mild and severe lupus groups. Consistent with DESeq findings, *Clostridium saudiense*, *Pseudoflavonifractor phocaeensis*, and *Intestinimonas butyriciproducens* were enriched in the severe lupus group. In contrast, *Kineothrix alysoides*, *Lactobacillus johnsonii*, and *Ruminiclostridium cellulolyticum* were enriched in the mild group. LEfSe analysis confirmed the predominance of Firmicutes in the severe group and *Lactobacillales* in the mild group.

A strong overlap between the DESeq and LEfSe analyses was observed, with taxa such as *Clostridium saudiense*, *Pseudoflavonifractor phocaeensis*, and *Intestinimonas butyriciproducens* consistently enriched in the severe lupus group. Conversely, *Kineothrix alysoides*, *Lactobacillus johnsonii*, and *Ruminiclostridium cellulolyticum* were associated with the mild lupus group. These overlapping taxa are visualized in Figure 4C, which graphically represents the dominant species shared between the two analyses.

Importantly, *Clostridium saudiense* emerged as the only taxon consistently enriched in the severe lupus group across both GI and fecal microbiota analyses, suggesting its critical role in lupus progression. However, the GI tract analysis revealed additional taxa (*Pseudoflavonifractor phocaeensis* and *Intestinimonas butyriciproducens*) not detected in fecal samples, highlighting the greater sensitivity and specificity of GI tract microbiota profiling in identifying disease-associated microbial shifts. The relative abundance (%) of these dominant microbial species identified in the stool and gastrointestinal analyses across the lupus severity groups is presented in Table 2, further demonstrating the compositional differences between the mild and severe groups. Additionally, the analysis of microbial composition based on read counts was consistent with the results obtained from relative abundance (%), confirming the reliability of the findings (Appendix A). Additionally, we performed DESeq and LEfSe analyses using read counts, and the results were consistent with those obtained using abundance values (Appendix A).

Figure 4C display correlation analyses, illustrating the relationship between the relative abundance of key microbial species and lupus severity. Strong positive correlations (R^2^ values) were observed for *Clostridium saudiense*, *Pseudoflavonifractor phocaeensis*, and *Intestinimonas butyriciproducens*, reinforcing their enrichment in the severe group. Notably, the R^2^ values for GI tract analysis were consistently higher compared to fecal analysis, indicating that GI microbiota composition is more closely associated with lupus severity and reflects disease progression more accurately. Conversely, *Kineothrix alysoides*, *Lactobacillus johnsonii*, and *Ruminiclostridium cellulolyticum* exhibited negative correlations, indicating their higher abundance in the mild lupus group.

Collectively, Figure 4 demonstrates the progressive shifts in GI tract microbiota composition with lupus severity. The overlap of *Clostridium saudiense* across both fecal and GI tract analyses underscores its significance as a potential biomarker for severe lupus. However, the identification of additional taxa unique to the GI tract, along with the higher R^2^ values observed, suggests that analyzing the GI microbiota provides a more comprehensive and accurate representation of disease-associated microbial changes compared to fecal analyses.

## 3. Discussion

In this study, we demonstrate that alterations in gut microbiome composition are closely associated with lupus severity in the MRL/lpr mouse model, suggesting that specific gut microbes may influence disease progression by modulating autoantibody production and renal impairment. Mice were grouped based on IgG, IgG2a, and ACR levels, which are well-established biomarkers commonly used to assess lupus severity and disease progression [14].

IgG plays a central role in lupus pathogenesis, serving as the primary autoantibody isotype responsible for immune complex formation and deposition in tissues. Elevated IgG levels reflect increased B cell activation and the production of autoantibodies targeting nuclear antigens, leading to inflammation and organ damage, particularly in the kidneys [15,16]. In lupus nephritis, IgG-containing immune complexes are deposited in the glomeruli, triggering complement activation and promoting inflammatory cascades that drive renal impairment [17]. This makes total IgG a key indicator of systemic autoimmunity and disease severity in lupus.

Similarly, IgG2a, a subclass of IgG, is closely linked to the severity of lupus manifestations. High levels of IgG2a reflect enhanced Th1-driven immune responses, contributing to more aggressive autoantibody production and increased complement fixation, exacerbating tissue damage and nephritis [18]. Elevated IgG2a levels are indicative of a shift towards pro-inflammatory pathways, further reinforcing immune complex deposition and chronic inflammation.

ACR serves as a clinical marker of renal involvement, a frequent and severe complication of lupus, with elevated urinary albumin excretion signifying glomerular damage and compromised kidney function [19]. The observed increase in IgG, IgG2a, and ACR levels in severe lupus mice may reflect heightened T cell activation or broader immune dysregulation [20]. This observation aligns with previous studies suggesting that autoantibody production and renal impairment are linked to immune responses involving multiple cell populations [21,22]. Despite being the same age (16 weeks), mice displayed varying degrees of lupus severity, reinforcing the heterogeneity of disease progression and highlighting the importance of these biomarkers to stratify severity.

Interestingly, we found that these biomarkers can be significantly influenced by specific gut microbiota. Specifically, we identified a reduction in the abundance of *Ruminiclostridium cellulolyticum*, *Lactobacillus johnsonii*, and *Kineothrix alysoides* in severe lupus mice compared to mild cases. These bacteria are known producers of butyrate and lactate, metabolites essential for maintaining intestinal and systemic immune homeostasis [22,23]. The depletion of these bacteria may lead to decreased butyrate and lactate levels, impairing FOXP3 expression and contributing to regulatory T cell (Treg) dysfunction [24,25]. This is particularly relevant as butyrate and lactate promote anti-inflammatory environments and are crucial for maintaining Treg populations [22,24]. A decline in these metabolites may exacerbate systemic inflammation, thereby enhancing IgG2a production and ACR elevation.

Conversely, severe lupus mice exhibited a pronounced increase in the abundance of *Clostridium saudiense*, *Turicibacter sanguinis*, *Ligilactobacillus animalis*, *Pseudoflavonifractor phocaeensis*, and *Intestinimonas butyriciproducens*, indicating a clear state of gut dysbiosis that may actively drive pro-inflammatory pathways and exacerbate disease severity. This dysbiosis is characterized by the enrichment of microbial species that may promote autoimmune activation and inflammatory responses, potentially accelerating lupus progression. Although direct evidence linking these specific species to lupus remains limited, their enrichment aligns with patterns observed in other autoimmune diseases, such as rheumatoid arthritis and multiple sclerosis, where gut dysbiosis may contribute to systemic inflammation and disease flares [26,27,28].

Notably, genera such as Clostridium and Turicibacter have been implicated in promoting Th17 differentiation and IL-17 production—key drivers of autoimmunity and tissue inflammation [29]. The Th17 pathway is critical in lupus pathogenesis, contributing to the breakdown of self-tolerance, autoantibody production, and subsequent organ damage [30]. Thus, the observed microbial shifts in severe lupus mice may reflect more than a byproduct of inflammation—they likely play a direct role in fostering immune dysregulation and sustaining the inflammatory environment required for disease exacerbation.

Our study provides the first evidence that microbial species such as *Clostridium saudiense*, *Turicibacter sanguinis*, *Ligilactobacillus animalis*, *Pseudoflavonifractor phocaeensis*, and *Intestinimonas butyriciproducens* may act as drivers of lupus severity, bridging the gap between gut dysbiosis and systemic autoimmune progression. Their consistent presence in severe lupus mice highlights a potential mechanistic link between gut microbial composition and heightened immunological activity. This reinforces the hypothesis that specific gut microbes exacerbate lupus by promoting pro-inflammatory Th17 pathways, increasing IL-6 production, and interfering with Treg function.

Equally important is the significant reduction in beneficial bacteria, including *Ruminiclostridium cellulolyticum*, *Lactobacillus johnsonii*, and *Kineothrix alysoides*, observed in severe lupus mice. These microbes are known producers of butyrate and lactate—short-chain fatty acids (SCFAs) essential for maintaining intestinal barrier integrity and modulating immune responses [12,31]. SCFAs promote the differentiation and stability of FOXP3+ regulatory T cells (Tregs), which play a critical role in preventing autoimmunity by suppressing excessive immune activation. The depletion of these beneficial bacteria may lead to reduced SCFA production, impairing Treg function and creating an imbalance that favors pro-inflammatory responses, further enhancing IgG2a production and renal damage.

This dual pattern of dysbiosis—marked by the enrichment of pro-inflammatory bacteria and the depletion of immunoregulatory microbes—represents a hallmark of lupus pathogenesis. The convergence of these microbial shifts creates a permissive environment for chronic inflammation, autoantibody production, and progressive organ damage. These findings align with previous reports highlighting the role of gut microbiota in modulating systemic lupus erythematosus by influencing cytokine profiles and T cell polarization [32,33].

Additionally, while this study primarily focused on how lupus severity affects gut microbiota composition, it is worth considering the reciprocal relationship wherein gut microbiota may actively influence lupus progression. Targeting pro-inflammatory taxa such as *Clostridium saudiense* or *Pseudoflavonifractor phocaeensis* with antibiotics or supplementing beneficial SCFA-producing microbes such as *Ruminiclostridium cellulolyticum* through probiotics could provide novel therapeutic strategies. Similarly, conventional treatments, such as steroids or immunosuppressive drugs, may influence both the systemic immune response and gut microbiota composition. These treatments could indirectly improve gut microbial balance by reducing systemic inflammation, which is closely linked to dysbiosis [34,35].

Conversely, restoring gut microbiota homeostasis through microbiota-targeted therapies, such as probiotics or prebiotics, may have the potential to complement conventional approaches by mitigating renal impairment and systemic lupus symptoms. While this study does not directly address the interaction between these treatments and gut microbiota, future studies exploring these multifaceted relationships will help elucidate the bidirectional interactions between gut microbiota and lupus severity. Such research may open avenues for microbiome-targeted therapies in autoimmune diseases, offering a more comprehensive approach to managing systemic lupus erythematosus [34,35].

Our findings underscore the importance of IgG, IgG2a, and ACR as essential biomarkers for assessing lupus severity, while also revealing a strong association between these immunological markers and gut microbiota composition. The identification of *Clostridium saudiense*, *Turicibacter sanguinis*, and other microbial species as potential contributors to disease progression provides valuable insights into the gut-immune axis in lupus pathogenesis.

While further studies are necessary to elucidate the precise mechanisms underlying these microbial shifts, our findings highlight the potential for gut microbiota to serve as both biomarkers and therapeutic targets for lupus. The modulation of gut microbiota through probiotic interventions or microbiome-targeted therapies could offer novel avenues to mitigate disease severity by restoring immune homeostasis and promoting beneficial microbial communities.

Overall, our findings underscore the importance of IgG2a and ACR as core biomarkers for assessing lupus severity while also revealing a strong association between these immunological markers and the gut microbiome. This study highlights the potential for gut microbiota to serve as adjunct markers influencing immunological indicators such as IgG2a and ACR. Further exploration of these microbial species may pave the way for developing microbiome-targeted interventions, potentially offering novel therapeutic avenues for lupus management and disease modification.

## 4. Materials and Methods

### 4.1. Animal

Ten 6-week-old and ten 16-week-old female MRL/lpr mice were purchased from Central Lab. Animal Inc. (Seoul, Korea). Five mice were randomly selected from the 6-week-old group. The 16-week-old mice were classified into moderate (n = 5) and severe (n = 5) groups based on disease severity. Microbiome analysis was conducted for all mice; however, data from two mice in the severe group were excluded due to errors encountered during the meta-analysis. All mice were individually housed at the Animal Facility of the Aging Science Department at the Korea Basic Science Institute (Gwangju, Republic of Korea) in a specific pathogen-free (SPF) facility with ad libitum access to sterilized food and water. The housing conditions were maintained at a temperature of 22 ± 1 °C and relative humidity of 40–50%, with a 12 h light/dark cycle. All experimental procedures adhered to institutional and ethical guidelines for animal research.

### 4.2. 16S rRNA Gene Sequencing

DNA was extracted from fecal or gastrointestinal bacterial samples using the phenol-chloroform-isoamyl alcohol method, as previously described [36,37,38,39]. Briefly, bead beating with a lysis buffer was performed to break the bacterial cell walls, followed by organic extraction to isolate the DNA. The DNA was then precipitated with ethanol, washed, dried, and resuspended in a suitable buffer.

The DNA concentration and purity were measured using a BioSpec-nano spectrophotometer (Shimadzu Biotech, Kyoto, Japan), and gel electrophoresis was used to check the DNA integrity.

Metagenomic sequencing was conducted by a commercial service provider (ebiogen, Inc., South Korea) using a next-generation sequencing protocol compatible with the Illumina platform. Specific regions of the 16S rRNA gene (V3–V4 primers) were amplified, and unique identifiers and sequencing adapters were added. The libraries were normalized, pooled, and sequenced using the Illumina platform. Data analysis included error correction, filtering, and removal of chimeric sequences and singletons. Metabolomics raw data have been uploaded to the figshare database and can be accessed at https://doi.org/10.6084/m9.figshare.28106627.v1 (accessed on 30 December 2024). Sequence data generated in this study was uploaded to the NCBI SRA database. It can be accessed via the accession number PRJNA1146986.

### 4.3. Characterizing Microbiome Diversity and Abundance Through α-Diversity Analysis

Bioinformatic tools, as previously described [36,37,38,39], were utilized to assess the core taxa and analyze their relative abundance across samples (α-diversity). The data were processed and normalized using software packages such as phyloseq and MetagenomeSeq, employing Cumulative Sum Scaling (CSS). Taxonomic abundances were calculated, and classifications were grouped for visualization. For clarity, taxa with relative abundance below 5% (excluding kingdom and class levels) were grouped into an “Others” category.

### 4.4. Characterizing Microbial Differences with β-Diversity Analysis

The β-diversity was assessed using Bray–Curtis dissimilarity based on log-transformed OTU data, as outlined in previous studies [36,37,38,39]. Non-metric multidimensional scaling (NMDS) was then applied using the ‘MetaMDS’ function from the ‘vegan’ package. NMDS helps to reduce the dimensionality of the data while preserving the most relevant information regarding the relationships between samples. This analysis was conducted on the Bray–Curtis dissimilarity matrix to explore the similarities and differences among the samples.

### 4.5. Visualizing Microbial Relationships Through Heatmaps and Phylogenetic Trees

Heatmaps and clustering analyses were created using the Heatplus package (version 2.30.0) from Bioconductor and the ‘vegan’ package in R, as described in previous studies [36,37,38,39]. These analyses employed the relative abundance values of all OTUs or the most abundant core OTUs across the samples. For clustering and heatmap generation, average linkage hierarchical clustering was applied, and Bray–Curtis dissimilarity was used for distance measurement. Prior to heatmap construction, a 5% prevalence threshold was applied to filter out less abundant taxa, ensuring that only the most abundant classification groups were included.

To visualize taxonomic composition across samples, phylogenetic trees were generated from the raw sequence data without prior filtering, following the methods outlined earlier [36,37,38,39]. Specifically, classification groups that could not be resolved to the species level were reclassified based on NCBI accession numbers using the Taxonomizr package (version 0.5.3) in R. The 16S rRNA sequences for each group were then aligned using ClustalW with default settings. The alignments were used to construct maximum likelihood phylogenetic trees with 500 bootstrap replicates, which were generated in MEGAX. All trees were visualized using iTOL.

### 4.6. Co-Occurrence Network Mapping of Microbial Communities

Co-occurrence relationships between microbial taxa were assessed using a co-abundance network approach [36,37,38,39]. Normalized abundance data were processed with tools like CoNet to detect significant co-occurrences across samples. Various similarity metrics were applied to evaluate these co-occurrences, and only relationships deemed significant by at least two metrics were included in the final network. The resulting network was then analyzed to identify potential clusters or communities of co-occurring taxa using established algorithms.

### 4.7. Microbial Abundance Analysis

Differential abundance analysis was performed following previously established protocols [36,37,38,39]. To evaluate changes in the taxonomic composition of gut microbiota across different muscle strength categories, we applied DESeq2 (version 1.24.0) and LEfSe (version 1.1.2). Taxonomic groups that were present in less than 1% of the samples were excluded from the analysis.

### 4.8. Enzyme-Linked Immunosorbent Assay (ELISA)

Blood was collected from the orbital sinus, and serum samples were stored at −20 °C until analysis. Serum IgG and IgG2a levels were measured using an ELISA kit (Condrex #3031 and Bethyl Laboratories #E99-107, respectively). Absorbance was determined with a Sunrise ELISA microplate reader (Tecan, Mannedorf, Switzerland). Statistical analysis was performed using Student’s *t*-test (SPSS v29, Chicago, IL, USA), and data were presented as mean ± standard error of the mean (S.E.M). Statistical significance was defined as *p* < 0.1.

### 4.9. Urine Albumin and Creatinine Assays

Urine samples were collected in aseptic microtubes. Albumin and creatinine concentrations were quantified using a mouse albumin ELISA kit (Bethyl Laboratories #E99-134) and a creatinine assay kit (R&D Systems, Minneapolis, MN, USA, #KGE005), following the manufacturer’s guidelines. Statistical analysis was performed using Student’s *t*-test (SPSS v29, Chicago, IL, USA), and results were expressed as mean ± standard error of the mean (S.E.M). Statistical significance was set at *p* < 0.1.

### 4.10. Data and Statistical Analyses

Data analysis was carried out using previously described bioinformatic tools [36,37,38,39]. Briefly, bacterial taxa were classified using a Naïve Bayes classifier, referencing the SILVA V3–V4 region database (https://www.arb-silva.de/). Denoising was performed with default parameters before taxonomic classification. Diversity metrics were computed using the q2-diversity software (R version 3.3.3), with settings that included a minimum sequencing quality score threshold of 20 and a rarefaction depth of 11,510. To assess differences in alpha and beta diversity, non-parametric tests were applied. Kruskal–Wallis and PERMANOVA tests were used to evaluate alpha and beta diversity, respectively. The Adonis test with multiple permutations was used to assess the effect size on Bray–Curtis dissimilarities. The Benjamani–Hochberg method was used to adjust *p*-values, with significance determined at *p* < 0.05.

### 4.11. Ethics Approval

All the experimental procedures complied with the ARRIVE guidelines, and the Institutional Animal Care and Use Committee of the Korea Basic Science Institute approved the animal protocols (KBSI-IACUC-23-36).

## 5. Conclusions

This study demonstrates that alterations in gut microbiota are associated with lupus severity in MRL/lpr mice, suggesting that changes in specific microbial populations may influence immune dysregulation. These microbial shifts correlate with biomarkers reflecting lupus severity, such as total IgG, IgG2a, and urine ACR levels. Our findings provide the first evidence identifying specific microbial species that may contribute to lupus severity, highlighting their potential as adjunct biomarkers and therapeutic targets for lupus. By linking distinct microbial alterations to elevated immunoglobulin levels and renal impairment, this study underscores the importance of the gut microbiome in shaping lupus progression and severity. Future microbiota-targeted interventions aimed at restoring microbial balance may offer promising therapeutic avenues to mitigate disease progression and improve patient outcomes.

## Figures and Tables

**Figure 1 ijms-26-01006-f001:**
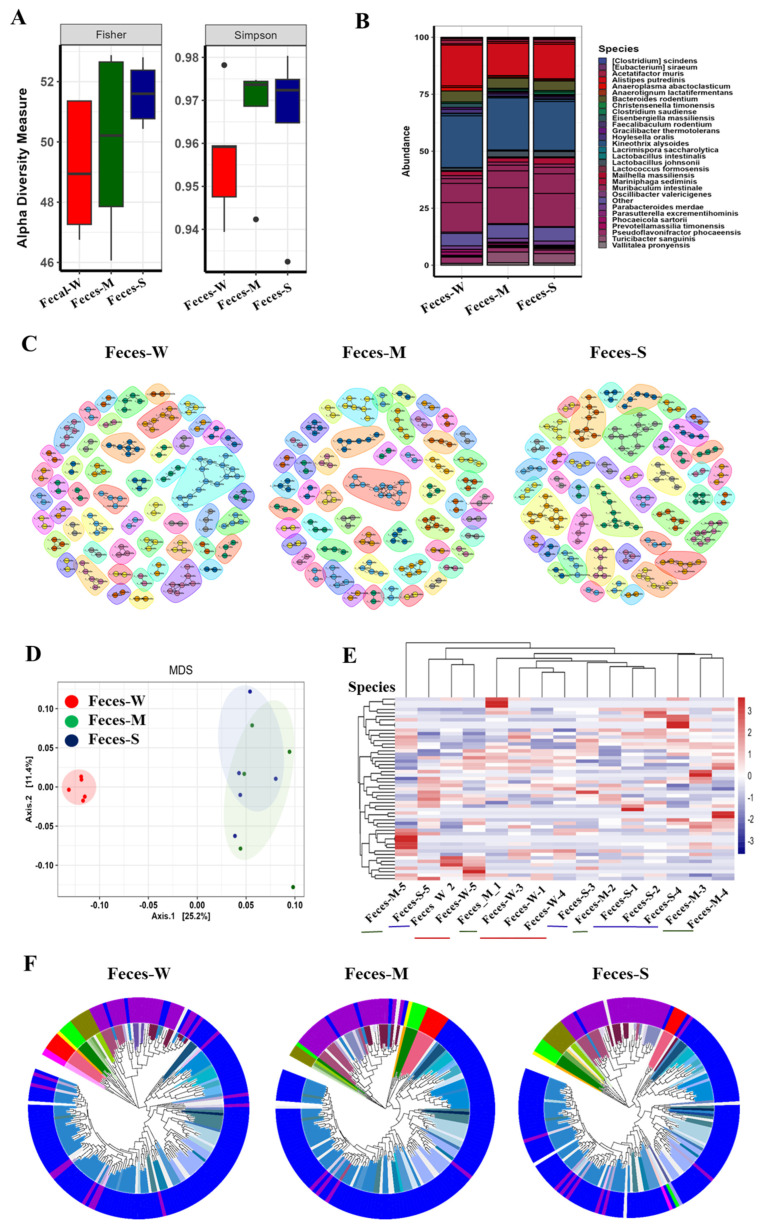
Changes in fecal microbiota composition associated with lupus severity in MRL/lpr mice. (**A**) Alpha diversity measures (Fisher and Simpson indices) comparing gut microbiota diversity between mild (M), severe (S), and wild-type (W) mice. (**B**) Taxonomic composition of gut microbiota at the species level across different groups. (**C**) Co-occurrence network analysis illustrating microbial community structures in wild-type (**left**), mild lupus (**middle**), and severe lupus (**right**) groups. (**D**) Multidimensional scaling (MDS) plot showing beta diversity and clustering of gut microbiota profiles. (**E**) Heatmap of differentially abundant microbial species across the groups. (**F**) Circular bar plots representing taxonomic abundance profiles of gut microbiota at different taxonomic levels (phylum to species) for wild-type (**left**), mild (**middle**), and severe (**right**) lupus mice.

**Figure 2 ijms-26-01006-f002:**
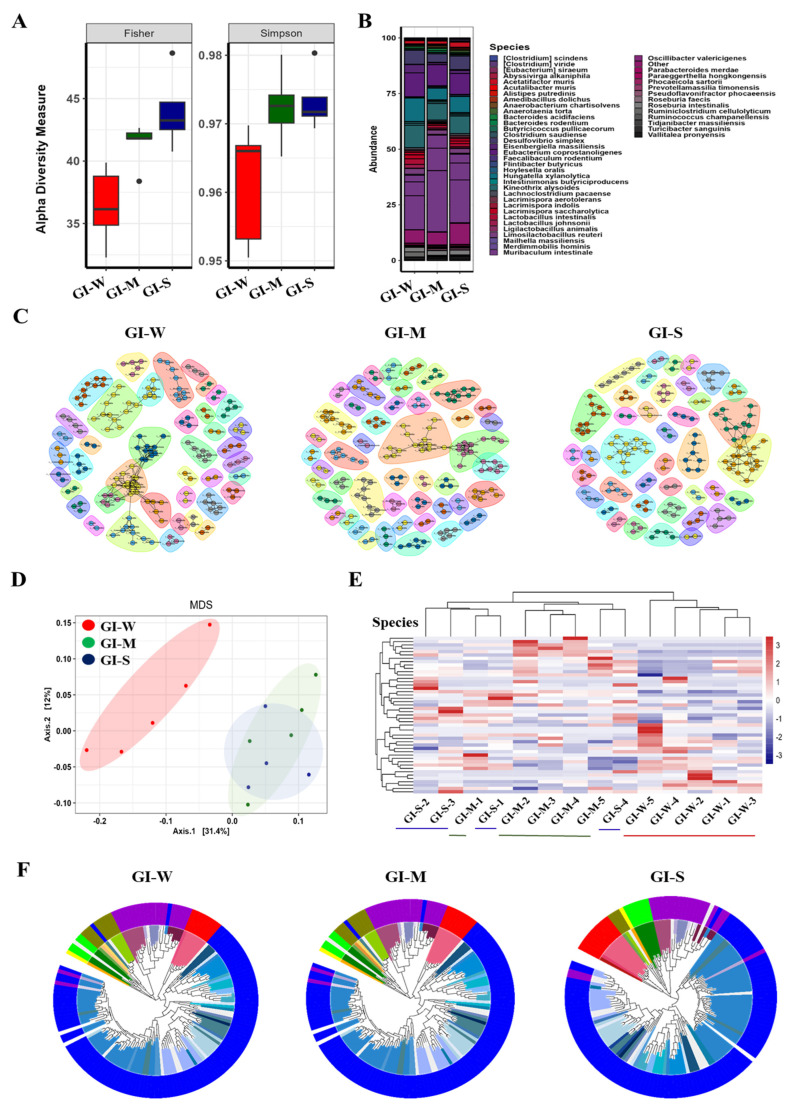
Changes in GI microbiota composition associated with lupus severity in MRL/lpr mice. (**A**) Alpha diversity measures (Fisher and Simpson indices) comparing gut microbiota diversity between mild (M), severe (S), and wild-type (W) mice. (**B**) Taxonomic composition of gut microbiota at the species level across different groups. (**C**) Co-occurrence network analysis illustrating microbial community structures in wild-type (**left**), mild lupus (**middle**), and severe lupus (**right**) groups. (**D**) Multidimensional scaling (MDS) plot showing beta diversity and clustering of gut microbiota profiles. (**E**) Heatmap of differentially abundant microbial species across the groups. (**F**) Circular bar plots representing taxonomic abundance profiles of gut microbiota at different taxonomic levels (phylum to species) for wild-type (**left**), mild (**middle**), and severe (**right**) lupus mice.

**Figure 3 ijms-26-01006-f003:**
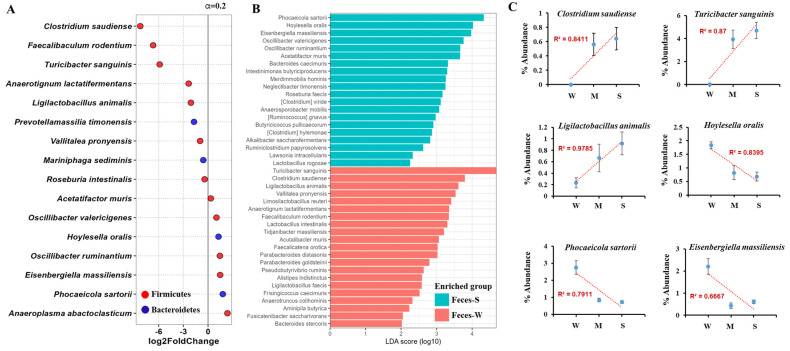
Differential analysis of fecal gut microbiota in wild-type (W) and severe lupus (S) MRL/lpr mice. (**A**) Volcano plot showing differentially abundant fecal microbial species between severe lupus (Feces_S) and wild-type (Feces_W) mice. Species from the phylum Firmicutes are marked in red, and species from Bacteroidetes are in blue. Positive values indicate species enriched in severe lupus mice, while negative values indicate those enriched in wild-type mice. (**B**) Linear discriminant analysis (LDA) effect size (LEfSe) plot showing microbial species significantly enriched in fecal samples from severe lupus (Feces_S, blue) and wild-type (Feces_W, red) mice. Species with higher LDA scores indicate greater contributions to the group’s microbial profile. (**C**) Correlation analysis between specific microbial species and lupus-associated biomarkers. Scatter plots show the relationship between microbial abundance and biomarkers, with linear regression lines and R^2^ values indicated for each correlation.

**Figure 4 ijms-26-01006-f004:**
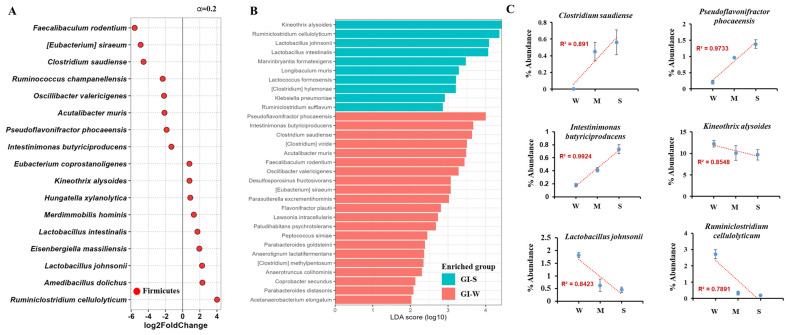
Differential analysis of the GI microbiome of wild-type (W) and severe lupus (S) MRL/lpr mice. (**A**) Volcano plot showing differentially abundant GI microbial species between severe lupus (GI_S) and wild-type (GI_W) mice. Species from the phylum Firmicutes are marked in red, and species from Bacteroidetes are in blue. Positive values indicate species enriched in severe lupus mice, while negative values indicate those enriched in wild-type mice. (**B**) Linear discriminant analysis (LDA) effect size (LEfSe) plot showing microbial species significantly enriched in GI samples from severe lupus (GI_S, blue) and wild-type (GI_W, red) mice. Species with higher LDA scores indicate greater contributions to the group’s microbial profile. (**C**) Correlation analysis between specific GI microbial species and lupus-associated biomarkers. Scatter plots show the relationship between microbial abundance and biomarkers, with linear regression lines and R^2^ values indicated for each correlation.

**Table 1 ijms-26-01006-t001:** Grouping of MRL/lpr mice according to immunological markers.

Group	IgG2a (μg/mL)	IgG (μg/mL)	Urine ACR (mg/g)
Lupus-weak	1013.70 ± 30.26	210.55 ± 44.92	44.51 ± 12.14
Lupus-medium	1691.75 ± 9.54	2069.26 ± 879.29	102.34 ± 20.25
Lupus-strong	1736.46 ± 21.34	3133.95 ± 649.04	461.16 ± 188.68

**Table 2 ijms-26-01006-t002:** Differentially abundant fecal and GI gut microbial species based on relative abundance in mild and severe lupus MRL/lpr mice.

Sample	Strain	Relative Abundance (%)		
		Lupus-Weak	Lupus-Medium	Lupus-Strong
Fecal	*Clostridium saudiense*	0.00	0.56	0.64
	*Turicibacter sanguinis*	0.01	3.93	4.71
	*Ligilactobacillus animalis*	0.23	0.66	0.91
	*Hoylesella oralis*	1.83	0.82	0.67
	*Phocaeicola sartorii*	2.74	0.83	0.72
	*Eisenbergiella massiliensis*	2.20	0.60	0.43
GI	*Clostridium saudiense*	0.00	0.45	0.56
	*Pseudoflavonifractor phocaeensis*	0.20	0.96	1.38
	*Intestinimonas butyriciproducens*	0.18	0.41	0.73
	*Ruminiclostridium cellulolyticum*	2.71	0.33	0.20
	*Lactobacillus johnsonii*	1.81	0.63	0.46
	*Kineothrix alysoides*	12.21	10.07	9.71

## Data Availability

Metabolomics raw data generated in this study have been uploaded to the Figshare database and can be accessed at https://doi.org/10.6084/m9.figshare.28106627.v1 (accessed on 30 December 2024). Sequence data generated in this study have been uploaded to the NCBI SRA database and are available via the accession number PRJNA1146986. All datasets analyzed or generated during this study are publicly available as stated above.

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
