# Peer review of "Changes in Gut Microbiota According to Disease Severity in a Lupus Mouse Model"

_ijms, 2025, doi:10.3390/ijms26031006_

Round 1
Reviewer 1 Report
Comments and Suggestions for Authors
See the file

Author Response
The attached document contains a more detailed answer.

Reviewer 2 Report
Comments and Suggestions for Authors
A manuscript submitted by Han and colleagues emphasizes the critical role of gut microbiota dysbiosis and immune dysregulation in lupus pathogenesis, with severe lupus linked to a reduction in beneficial microbes and an increase in pathogenic species. The study suggests that these microbial imbalances may drive disease progression by impairing regulatory T cell function and enhancing B and T cell activation, highlighting potential biomarkers and therapeutic targets for systemic lupus erythematosus (SLE). The manuscript is interesting and suitable for publication; however, the authors should address a few minor points for clarity:
- Please provide the full form and a one-sentence introduction of the MRL/lpr mouse model and CyTOF analysis in the abstract.
- Expand on the background of systemic lupus erythematosus (SLE) in the introduction section.
These clarifications will enhance the manuscript’s overall interpretation and accessibility.
Author Response
The attached document contains a more detailed answer
